# Preservation and degradation of ancient organic matter in mid-Miocene Antarctic permafrost

Marjolaine Verret<sup>1</sup>, Sebastian Naeher<sup>2,3</sup>, Denis Lacelle<sup>4</sup>, Catherine Ginnane<sup>3</sup>, Warren Dickinson<sup>5</sup>, Kevin Norton<sup>2</sup>, Jocelyn Turnbull<sup>3,6</sup>, Richard Levy<sup>3,5</sup>

- <sup>5</sup> Department of Arctic Geology, The University Centre in Svalbard, Longyearbyen, 9170, Svalbard and Jan Mayen.
  - <sup>2</sup>School of Geography, Environment and Earth Sciences, Victoria University of Wellington, Wellington, 6011, New Zealand.
  - <sup>3</sup>GNS Science, Lower Hutt, 5010, New Zealand.
  - <sup>4</sup>Department of Geography, Environment and Geomatics, University of Ottawa, Ottawa, K1N 6N5, Canada.
  - <sup>5</sup>Antarctic Research Centre, Victoria University of Wellington, Wellington, 6011, New Zealand.
- 10 <sup>6</sup>CIRES, University of Colorado at Boulder, USA.

Correspondence to: Marjolaine Verret (marjolainv@unis.no)

**Abstract.** The Antarctic environment is amongst the coldest and driest environments on Earth. The ultraxerous soils in the McMurdo Dry Valleys support exclusively microbial communities, however, 15 million years ago, a tundra ecosystem analogous to present-day southern Greenland occupied this region. The occurrence of ancient soil organic carbon combined with low accumulation of contemporary material makes it challenging to differentiate between ancient and modern organic processes. Here, we explore the additions of modern organic carbon, and the preservation and degradation of organics and lipid biomarkers, in a 1.4 m mid-Miocene age (~14.5-14.3 Ma) permafrost soil column from Friis Hills. The total organic carbon is low throughout the soils (< 1%wt). The near-surface (upper 35 cm) dry permafrost has lower C:N ratios, higher  $\delta^{13}$ C<sub>org</sub> values, higher proportion of branched fatty acids with an iso- and anteiso- configuration relative to *n*-fatty acids, lower phytol abundance and higher contributions of low-molecular weight homologues of n-alkanes, than the underlying icy permafrost, indicating higher contributions from bacteria-derived organic matter. Conversely, the icy permafrost contains higher molecular weight n-alkanes, n-fatty acids and n-alkanels, along with phytosterols (e.g., sitosterol, stigmasterol) and phytol (and its derivatives pristane and phytane) that are indicative of the contributions and preservation of higher-level plants. This implies that legacy mid-Miocene age carbon in the near-surface soils (c. 35 cm) has been prone to microbial organic matter degradation during times when the permafrost thawed, likely during relatively warm intervals through the late Neogene ( $\sim$ 6.0 Ma) and sporadically during the Holocene (<1%), when ground summer temperatures were  $\geq$ +2°C (based on brGDGTs temperature reconstructions). Conversely, lipid biomarkers found deeper in the permafrost have been preserved for millions of years. These results suggest that ancient organics preserved in permafrost could underpin significant ecological changes in the McMurdo Dry Valleys under the current warming climate.

#### 0 1 Introduction

The McMurdo Dry Valleys (MDV) are amongst the coldest and driest environments on Earth (e.g. Horowitz et al., 1972). This hyperarid polar desert can be divided into three microclimate zones (Marchant and Head III, 2007): (i) the coastal thaw (or

subxerous) zone (<400 m above sea level; a.s.l.) where mean daily summer air temperatures exceed 0°C and soil water can exist seasonally; (ii) the inland mixed (or *xerous*) zone where summer air temperatures may rise above 0°C for short periods and where soil water may be present periodically; and the *ultraxerous* or stable upland zone, where maximum air temperatures rarely exceed 0°C and snow is scarce (<10 mm water equivalent per year). The MDV lack vascular plants, and the subxerous zone contains edaphic communities with sparse cryptogamic vegetation (mosses and lichens), low diversity of invertebrates, and heterotrophic soil organisms, as well as endolithic communities of phototrophic and heterotrophic organisms in sandstone outcrops and other lithic substrates (e.g. Bargagli et al., 1999; Barrett et al., 2007; Cary et al., 2010; Freckman and Virginia, 1997; Horowitz et al., 1972; Moorhead et al., 2002). By contrast, the ultraxerous zone lacks all types of edaphic vegetation and appears to only support endolithic organisms in sandstone outcrops (Friedmann, 1982). Heterotrophic microbial communities are present in soils but at very low metabolic levels or in a dormant state (Goordial et al., 2016; Tamppari et al., 2012). Similar conclusions were drawn from the Shackleton Glacier region, a region situated further inland in the Transantarctic Mountains, where cultivation-dependent, cultivation-independent and metabolic assays were unable to detect viable microbial life (Dragone et al., 2021). However, in places in University Valley (Fig. 1a) where the ground surface raises above 0°C in summer, the permafrost contains about 5× more organic carbon ( $C_{org}$ ), and microbial activity has been indirectly inferred from C:N ratios,  $\delta^{13}C_{caco3}$  (Faucher et al., 2017).

40

The studies about the source and cycling of C<sub>org</sub> in University Valley were from permafrost cores that represents about 152 kyr of sediment accumulation (Lacelle et al., 2013). At Friis Hills (Fig. 1a), the permafrost consists of early to middle Miocene age sediments (Chorley et al., 2023; Lewis and Ashworth, 2016), and is potentially the oldest permafrost on Earth (Verret et al., 2021). During the mid-Miocene (~20-14.6 Ma), the region harboured a tundra environment (Lewis and Ashworth, 2016; Chorley et al., 2023; Lewis et al., 2008) analogous to that found in southern Greenland today (including vascular plants). Since there have been limited periods of melt since the mid-Miocene (Verret et al., 2023), the site offers the opportunity to analyse the preservation and degradation of organic carbon in permafrost over millions of years. Lipid biomarkers are among the most stable organic molecules and can be preserved over long time periods (e.g. Naeher et al., 2022; Schouten et al., 2013; Castañeda and Schouten, 2011; Duncan et al., 2019). Low-molecular weight organic compounds (e.g. sugars and amino acids) and lipids with double bonds or polar functional groups (e.g. fatty acids and alcohols) are typically susceptible to microbial decomposition. However, other lipid biomarkers, such as apolar, saturated hydrocarbons (e.g. alkanes), isoprenoids (e.g. phytane) and cyclic compounds (i.e. hopanes or steranes), are refractory compounds formed during diagenesis (Peters et al., 2004b) and may be preserved on geological timescales (e.g. Eigenbrode, 2008). Over large timescales, both biotic and abiotic degradation occurred at Friis Hills. Abiotic degradation would result in the organic material near the surface to be better preserved (due to the principle of superposition). However, in a permafrost environment, biotic degradation, which is restricted to the active layer (layer that thaws in freezes on an annual basis), would overprint the abiotic degradation and would be more important near the surface: incorporating both contemporary and ancient organic carbon into the system (Kusch et al., 2021).

As such, the distribution and occurrence of biomarkers in the Friis Hills Miocene age sediments can provide insight about depositional conditions as well as paleoenvironmental changes post deposition (e.g. Duncan et al., 2019).

In this study, we investigate the source of ancient carbon reservoirs, degradation and potential overprint of modern organic carbon sources by combining bulk organic carbon and nitrogen analyses (TOC, total N,  $\delta^{13}C_{org}$ ), with lipid biomarker indicators, ramped pyrolysis-gas chromatography-mass spectrometry (Py-GC-MS), and radiocarbon dating. We also use bacterial branched glycerol dialkyl glycerol tetraether (brGDGT) lipids to infer threshold temperatures required to unlock organic carbon stored within the permafrost. The exposure of ancient organic carbon to modern degradation at Friis Hills provides valuable information on the present-day organic processes, which can be juxtaposed to the distinct organic signature of the tundra environment that dominated the site during sediment deposition. This allows us to assess the degradation of ancient organic carbon pools, soil carbon mineralization and cycling in the context of a warming Antarctic climate.

## 2 Study Site

75

The Friis Hills (77°45'S, 161°30'E, 1200-1500 m a.s.l.) are a 12 km wide inselberg situated at the head of Taylor Valley, ca. 50 km from the Ross Sea coast (Fig. 1a). The Friis Hills are situated within the ultraxerous zone (Marchant and Head III, 2007), where modern climatic conditions are extremely cold and arid, and microbial life is limited. The mean annual air temperature is -22°C, with an average reaching -13°C during the summer months (Friis Hills Meteorological Station 2011-2015; Doran and Fountain, 2016). The area receives less than 10 mm snow water equivalent of precipitation per year, most of which is displaced by strong katabatic winds (yearly average wind speed at Friis Hills is 4.7 m s<sup>-1</sup>; Bliss et al., 2011). The local geology consists primarily of Ordovician and Jurassic age intrusive igneous rocks and Devonian sedimentary rocks of the Beacon Supergroup (Allibone et al., 1993; Lewis and Ashworth, 2016; Cox et al., 2012). A ca. 80 m thick sequence of glacial drifts interbedded with lacustrine sediments record multiple episodes of advance and retreat of local alpine glaciers and intervals of ice sheet over-riding (Chorley et al., 2023; Lewis and Ashworth, 2016).

Figure 1 a. Location map of Friis Hills (FH) within the McMurdo Dry Valleys of Antarctica. Contour lines at 200 m intervals. b. sedimentary log of core FHDP2C from Friis Hills with the upper 140 cm studied in detail in this paper, where colours reflect Munsell colours of each unit. c. organic carbon concentration, expressed in wt%, d.  $\delta^{13}$ C, e. C:N ratio and f.  $\delta^{13}$ Corg and C:N biplot with bulk sediment measurements from three different units in FHDP2C compared to ultraxerous environments; University Valley and Mackay Glacier (Faucher et al., 2017; Van Goethem et al., 2020), subxerous environments; Miers, Garwood, Taylor and Victoria Valleys (Barrett et al., 2007; Hopkins et al., 2009), Antarctic lake/stream mats (Hopkins et al., 2009; Lawson et al., 2004), Arctic soils (Haugk et al., 2022; Osburn et al., 2019) and signature for C3 plants (Meyers, 1994). Unit separation was established by hierarchical clustering.

Major paleoenvironmental changes at Friis Hills can be divided into four distinct periods: (1) the early to mid-Miocene, a period when tundra persisted in the Friis Hills and throughout many locations across the Transantarctic Mountains (~20–14.6 Ma; Lewis et al., 2008; Duncan et al., 2019), (2) a period when tundra vegetation disappeared from high elevation localities in the MDV (~13.8 to 12.5 Ma; Lewis et al., 2008; Lewis and Ashworth, 2016; Chorley et al., 2023); (3) the mid-late Miocene (12.5–6.0 Ma), a period through which high elevations became progressively drier, culminating in hyper-arid conditions (Verret et al., 2023); and (4) the late Neogene and Quaternary (6.0 Ma–present) period, when the high elevations remained frozen and hyper-arid, even during relatively warm interglacial conditions. Low erosion rates (on the order of 0.1 m Myr<sup>-1</sup>) associated with a relatively cold and dry environment enhanced the preservation of the sediment record through the Neogene, while exposing the mid-Miocene organic carbon to surficial processes (Verret et al., 2023).

The Friis Hills Drilling Project (FHDP) recovered a series of sediment cores from three sites (1, 2, and 3) during the 2016-17 Antarctic Field season (Chorley et al., 2023). Three cores (A, B, and C) were drilled at Site 2 (77°45'17.028"S, 161°27'20.632"E, 1244 m a.s.l.) and this study focuses on the uppermost 140 cm of the ~ 5 m-long core 'C' (FHDP2C; Fig. 1a). The ice table (i.e. the boundary between the dry and the ice-cemented permafrost) occurs at 35 cm below the top of the core (Figs 1b & S1). The detailed age model of the FHDP cores is presented in Chorley et al. (2023). The lower section of core FHDP2C, which directly correlates with core FHDP2A, is constrained by <sup>40</sup>Ar/<sup>39</sup>Ar dating of a tephra at 5.13-5.18 m depth (14.4 Ma; Fig 1b). The most parsimonious solution for the normal chronology above the ash is that it is C5AD (~14.6 to 14.2 Ma), corresponding to the mid-Miocene climate transition (MMCT; Flower and Kennett, 1994; Shevenell et al., 2004). Furthermore, the analyzed sediments are from the upper part of sequence 13 from the Friis Hills composite presented in (Chorley et al., 2023). This sequence was likely deposited during a single precession cycle between 14.36 and 14.38 Ma (see Fig. 11 in Chorley et al., 2023) although we acknowledge that it may have been deposited during alternate precession cycles between ~14.4 and 14.2 Ma. Independent cosmogenic surface exposure dating at Friis Hills also suggests a minimum age of

110

115

the surface of 11 Ma (Valletta et al., 2015).

The FHDP2C core includes two sedimentary cycles comprising two glacial (i.e., matrix-supported diamicts attributed to subglacial traction tills) and two interglacial facies (i.e., muddy-sand units attributed to proglacial ponds). An assemblage of organic macrofragments such as lichens, liverworts, mosses, dicots, grasses and sedges, along with cuticles belonging to the *Nothofagaceae* family were observed within core samples (Chorley et al., 2023). Similar plant assemblages are widespread in the modern southern Greenland or Svalbard in the Arctic (e.g. Berke et al., 2019). Tundra macrofossils are found in all sedimentary facies but are most prevalent in the finer grained units, suggesting that higher plants were relatively abundant across the Friis Hills during the interglacial periods. These data indicate that the mid-Miocene climate at relatively high elevations in the MDV region of the Transantarctic Mountains remained warm and wet enough to support growth of higher plants (Chorley et al., 2023), after which tundra vegetation disappeared from these elevations (Lewis et al., 2008).

## 3 Methods

## 130 3.1 Core description and sampling

Lithological descriptions were undertaken in the field and cross-checked on the frozen core prior to sampling at GNS Science's National Ice Core Facility, Lower Hutt, New Zealand. The core was sub-sampled at ~5 cm intervals using a tile cutter with a 2.2 mm thick blade and samples were placed into polyethylene bags to thaw before being dried at 80°C for 24 hours (plastic contamination was not found in the samples). The surficial dry permafrost layer was sampled in the field at 5 cm intervals. A total of 23 samples from the upper 140 cm of the core were analysed for bulk organic carbon and nitrogen and lipid biomarkers.

#### 3.2 Bulk carbon and nitrogen analysis

The total organic carbon (TOC) and total nitrogen (TN) were measured to determine: i) the soil organic carbon density (SOCd), and ii) if the C:N ratios in the bulk sediments follow the Redfield biological stoichiometry ratio (6:61; Redfield, 1934). The TOC and TN was measured using an Elemental VarioEl Cube instrument at the Jan Veizer Laboratory, University of Ottawa, Canada. The samples were first acidified with 10% HCl to remove the inorganic carbon and isolate the organic carbon fraction. 100 mg of sediment was analysed within tin capsules, along with 100 mg of tungstic oxide (WO<sub>3</sub>), a combustion catalyst and binder. Calibrated standards of Sulfanilic Acid were prepared in a range of weights and ran cyclically to ensure instrument precision, and approximately 20% of samples were analysed in duplicate with analytical precision of  $\pm$  0.1%.

The  $\delta^{13}C_{org}$  of the bulk samples was measured using a DeltaPlus Advantage instrument coupled with the ConFlo III interface to assess potential source and alteration of organic carbon in the sediments. The results are expressed in  $\delta$  notation, which represents the parts per thousand difference of  $^{13}C/^{12}C$  in per mil (‰) with respect to the Vienna Pee-Dee Belemnite (VPDB) standard. Analytical precision was  $\pm$  0.2‰.

## 3.3 Lipid biomarker analyses

Lipid biomarkers were analysed to differentiate between the mid-Miocene tundra carbon and subsequent microbial degradation. Lipid biomarkers were analysed in the Organic Geochemistry Laboratory at GNS Science, Lower Hutt, New Zealand, following Naeher et al. (2012; 2014) with some modifications. Homogenised sediment (~5–15g) was extracted 4× by ultrasonic extraction using dichloromethane (DCM):methanol (MeOH) (3:1, v:v). Total lipid extracts (TLEs) were saponified using 6% KOH in MeOH at 80°C for 3 hours. After the addition of ultrapure H<sub>2</sub>O, neutrals were extracted from the aquatic solution using *n*-hexane. Following acidification with 6M HCl to pH <2, fatty acids (FAs) were then recovered using *n*-hexane.

The neutral fraction was further separated into apolar and polar fractions by silica gel chromatography using *n*-hexane and DCM:MeOH (1:1, v: v), respectively. Prior to analysis by gas chromatography-mass spectrometry (GC-MS), an aliquot of the polar fraction was derivatized with BSTFA in pyridine at 80°C for 1 hour, whereas FA fractions were derivatized with BF<sub>3</sub> MeOH at 100°C for 2 hours to obtain fatty acid methyl esters (FAMEs).

The resulting lipid fractions were analysed by GC-MS on an Agilent 7890A GC System, equipped with an Agilent J&W DB-5ms capillary column [60 m × 0.25 mm inner diameter (i.d.) × 0.25 μm film thickness (f.t.)], and coupled to an Agilent 5975C inert MSD mass spectrometer. The oven was heated from 70°C (held for 1 min) to 100°C at 20°C min<sup>-1</sup>, then to 320°C at 4°C min<sup>-1</sup> and held at that temperature for 20 minutes. Helium was used as carrier gas with a constant flow of 1.0 ml min<sup>-1</sup>. Samples (1 μl) were injected splitless at an inlet temperature of 300°C. The MS was operated in electron impact ionisation mode at 70 eV using a source temperature of 230°C.

Lipid biomarkers were quantified relative to an internal standard (50  $\mu$ L added to the TLEs following solvent extraction; consisting of 110.8  $\mu$ g ml<sup>-1</sup> 5 $\alpha$ -cholestane, 118.8  $\mu$ g ml<sup>-1</sup> n-nonadecanoic acid and 116.8  $\mu$ g ml<sup>-1</sup> n-nonadecanol). Procedural blanks were also analysed to ensure data quality and absence of laboratory contamination.

Another aliquot of the polar fraction containing glycerol dialkyl glycerol tetraethers (GDGTs) were dissolved in n-170 hexane/isopropanol (99:1, v:v) and filtered with 0.45 µm PTFE filters prior to analysis by high performance liquid chromatography-atmospheric pressure chemical ionisation-mass spectrometry (HPLC-APCI-MS) as reported in Hopmans et al. (2016). These analyses were undertaken on an Agilent 1260 Infinity II Prime LC system coupled to an Agilent 6125B single quadrupole MS following the method of Hopmans et al. (2016). In brief, two UHPLC silica columns (Acquity BEH HILIC columns, 2.1 × 150 mm, 1.7 um; Waters) were used in series, fitted with a 2.1 × 5 mm 175 pre-column of the same material (Waters), and maintained at 30°C. GDGTs were eluted isocratically for 25 minutes with 82% A:18% B, followed by a linear gradient up to 35% B in 25 minutes, then a linear gradient to 100% B in 30 min. where A is n-hexane and B is n-hexane/isopropanol (9:1, v/v). Flow rate was 0.2 ml/min. The abundances of GDGTs were monitored using selective ion monitoring mode (SIM) with m/z 1302, 1300, 1298, 1296, 1294, 1292, 1050, 1048, 1046, 1036, 1034, 1032, 1022, 1020, 1018. Compounds were identified by comparing mass spectra and retention times 180 with those in the literature (e.g. Hopmans et al., 2016). The concentration of biomarkers is expressed in microgram per gram of TOC (ug  $g^{-1}$  TOC).

## 3.4 Lipid biomarker proxies and indices

The concentrations of the lipid biomarkers were used to calculate a series of indices to: i) differentiate between plant- and bacteria-derived organic carbon; and ii) define the level of degradation of the organic carbon.

## 3.4.1 Carbon preference index



The carbon preference index (CPI) is a parameter that quantifies the ratio of odd to even *n*-alkanes and ratio of even to odd *n*-fatty acids based on its carbon number and is an indicator of the degradability of organic carbon. Natural distributions or well-preserved *n*-alkane signatures are expected to show a predominance of odd-numbered *n*-alkanes and even numbered *n*-fatty acids corresponding to high CPI values. The CPI value decreases with ongoing alteration of organic carbon (Bray and Evans, 1961):

$$CPI_{25-33} = 0.5 \cdot \left(\frac{\sum odd \ C_{25-31}}{\sum even \ C_{26-32}} + \frac{\sum odd \ C_{27-33}}{\sum even \ C_{26-32}}\right) \tag{1}$$

A similar index was also calculated for fatty acids and n-alcohols (Meyers and Ishiwatari, 1993):

$$CPI_{22-28} = \frac{\sum even \, C_{22-26} + \sum even \, C_{24-28}}{2 \cdot \sum odd \, C_{23-27}} \tag{2}$$

## 3.4.2 Average chain length and dominant chain length

In general, cuticle waxes of terrestrial plants contain predominantly high-molecular weight *n*-alkanes (>*n*-C<sub>27</sub>; with leaf waxes containing mainly *n*-C<sub>31</sub> and *n*-C<sub>33</sub> alkanes) while mid molecular weight (*n*-C<sub>21</sub> to *n*-C<sub>25</sub>) alkanes are indicative of aquatic macrophytes, mosses (e.g. *Sphagnum* in peat) or lichen, and low-molecular weight (*n*-C<sub>12</sub> to *n*-C<sub>20</sub>) alkanes are common in algae and bacteria (e.g. Naafs et al., 2019; Killops and Killops, 2013; Zech et al., 2010). A similar relation can be extracted from even carbon numbered *n*-alkanols and *n*-fatty acids. The average chain length (ACL) of *n*-alkanes can be used to identify the main source of organic carbon and it can be useful to infer environmental changes in a particular ecosystem (e.g. Killops and Killops, 2013). The ACL was calculated for odd-chain *n*-alkanes with C<sub>27</sub> to C<sub>33</sub> following this equation (Poynter and Eglinton, 1990):

$$ACL_{C27-C33} = \frac{\sum i \cdot C_i}{\sum C_i} \tag{3}$$

where i is the carbon number and  $C_i$  is the concentration of n-alkane.

ACL was also calculated for even-chain FAME ( $C_{22}$  to  $C_{32}$ ) and *n*-alkanols ( $C_{22}$  to  $C_{28}$ ). We also present the dominant chain length ( $C_{max}$ ) for each biomarker class.

#### 3.4.3 Ratio of short-chain and long-chain *n*-alkanes

The ratio of short-chain *n*-alkanes (SC, < C<sub>23</sub>) and long-chain *n*-alkanes (LC,  $\ge$  C<sub>23</sub>) was calculated using the following equation:

$$SC:LC = \frac{\sum C_{C15} - C_{22}}{\sum C_{C23} - C_{33}}$$
 (4)

High values of SC:LC are indicative of an environment dominated by bacteria while low values are indicative of an environment dominated by higher plants and/or macrophyte waxes (e.g. Killops and Killops, 2013).

#### 3.4.4. Other indices derived from saturated hydrocarbons, isoprenoids and fatty acids

The ratio of iso- and anteiso-FA to *n*-FA was calculated for the chain lengths detected in the samples (C<sub>15</sub>-C<sub>17</sub>) and expressed with a ×100 factor for ease of comparison. Branched FA compounds are diagnostic indicators of bacteria (e.g. Kaneda, 1991).

The pristane to phytane ratio (Pr:Ph) is used as a redox indicator (Powell, 1988; Naeher and Grice, 2015) and higher values reflect oxygenated conditions, which may represent increased anaerobic microbial degradation.

Since the natural  $\beta\beta$  isomer in  $C_{30}$  hopane is mostly present in modern organic material, we use the  $C_{30}$   $\alpha\beta/(\alpha\beta+\beta\alpha+\beta\beta)$  hopane index to infer the relative input of new bacterial-derived organic material (with a ratio <0.5 representing input of recent organic matter; Peters et al., 2004a; Farrimond et al., 1998).

#### 3.4.5. brGDGT indices

The branched and isoprenoid tetraether (BIT) index, which differentiates between inputs from a terrestrial environment (BIT  $\approx 1$ ) and a marine environment (BIT  $\approx 0$ ) was calculated as follows (Hopmans et al., 2004):

$$BIT\ Index = \frac{(I+II+III)}{(I+II+III+IV)} \tag{7}$$

where I, II and III denote the relative abundances of brGDGTs and IV reflects the relative abundance of crenarchaeol.

Past summer soil temperatures (Raberg et al., 2024) were estimated using the MBT'<sub>5ME</sub> index (De Jonge et al., 2014) and the soil-specific, Bayesian temperature calibration BayMBT<sub>0</sub> of Crampton-Flood et al. (2020). The MBT'<sub>5ME</sub> index was calculated

as


$$MBT'_{5ME} = \frac{I_a + I_b + I_c}{III_a + II_a + II_b + II_c + I_a + I_b + I_c}$$
(8)

#### 3.6 Radiocarbon and Pyrolysis-GC-MS analysis

To determine if biogeochemical processes are relict or currently active at Friis Hills, one sample of dry sediment (2-C0-3; 10 cm depth) was measured for radiocarbon content at the Rafter Radiocarbon Laboratory, GNS Science. The bulk sediment sample (< 300 µm) was first acid-treated to remove carbonates and then combusted at 900°C for 4 hours in an evacuated, sealed quartz tube with cupric oxide and silver wire. The resulting CO<sub>2</sub> was graphitized by reduction with hydrogen over iron catalyst and measured by accelerator mass spectrometry (Turnbull et al., 2015; Zondervan et al., 2015). Radiocarbon activity is presented as F<sup>14</sup>C (fraction modern carbon; Donahue et al., 1990; Reimer et al., 2004) and conventional radiocarbon age (as described in Stuiver and Polach, 1977).

To differentiate between ancient carbon reservoirs, diagenetic degradation and potential overprint of modern organic carbon sources, ~2 g of dried, homogenized decalcified material from the same sample 2-C0-3 was prepared for thermochemically-partitioned radiocarbon analysis using ramped pyrolysis oxidation-accelerator mass spectrometry (RPO-AMS) radiocarbon dating as reported in Ginnane et al. (2024). In brief, the sediment sample was apportioned into CO<sub>2</sub> fractions according to thermochemical lability. CO<sub>2</sub> was evolved from sample 2-C0-3 by pyrolysis with a constant ramp of 5°C min<sup>-1</sup> from room temperature to 750°C and subsequent oxidation, with discrete fractions collected from 105°C at 259, 395, 449, 535 and 750°C, respectively. The obtained CO<sub>2</sub> for each split was recombusted at 500°C with cupric oxide and silver wire and then graphitised and measured in the same way as the bulk sediment.

The same sample was also analysed by full, rapid-ramp pyrolysis and then incremental, partitioned ramped Py-GC-MS analysis in the Organic Geochemistry Laboratory at GNS Science to characterize the composition and sources of the organic carbon at different temperatures (Ginnane et al., 2024). For this, 37–50 mg of the sample was pyrolyzed in deactivated stainless-steel

cups. For full, rapid-ramp pyrolysis, samples were pyrolyzed with a ramp of 100°C min<sup>-1</sup> from 100°C to the Py-GC-MS instrumental maximum of 650°C. For incremental, partitioned ramped-Py-GC-MS, sample splits were obtained similar to RPO-AMS analysis by utilizing thermal desorption mode with a ramp of 10°C min<sup>-1</sup> run from 100°C to 259°C as the first split, then the subsequent splits collected to their respective maximum temperatures (395, 449, 535 and 650°C). After each split collection, the sample was removed from the pyrolyser furnace and GC-MS analysis of this step was completed, before reinserting the same sample cup and heating to the next split maximum temperature. The front inlet of the GC over was set to 5:1 split and the GC oven programme started at 40°C (held 5 min isothermal), ramped to 300°C at 5 °C min<sup>-1</sup>, and then held isothermal for 5 min. All other GC-MS settings were the same as described in Section 3.3.

Compounds were identified from the Py-GC-MS and grouped into nine compound classes following existing literature interpretations of dominant organic matter sources (i.e. bacteria or plant). The sources of *n*-alkanes were distinguished based on molecular weight (as detailed in section 3.4.2.). Pyrroles, furans and phenols were attributed to a plant-derived organic matter as they are derived from pigments such as chlorophylls and polysaccharides (e.g. Keely, 2006). Polycyclic aromatic hydrocarbons, thiophenes, alkylbenzenes and other aromatics are undifferentiable compounds.

## 3.7 Statistical analysis

To simplify the data, we carried out hierarchical clustering of all 25 variables in R Studio 1.3.1056 using the *chclust* function in the package *rioja* with the method *coniss* (Juggins, 2020).

#### 4 Results



The hierarchal clustering analysis of all samples in the 1.4 m sediment core and their 25 variables, which included TOC, C:N ratio, δ¹³C₀rg, lipid abundance and main biomarker indices, grouped the samples by depth. Group 1 consists of the dry permafrost from 0 to 35 cm depth; group 2 consists of the icy permafrost from 38 to 71 cm; and group 3 was the icy permafrost from 93 to 140 cm (Fig. 1b). An outlier sample (classified in a separate group) was identified at 79 cm depth and not included in any unit. The results first describe the general trends found throughout the core, followed by the variables and indices that are statistically different between the units.

#### 4.1 General biomarker trends

We identified the following components within the 1.4m core: n-alkanes (n-C<sub>15</sub> to n-C<sub>34</sub>), isoprenoids, hopanoids, n-fatty acids (n-C<sub>12</sub> to n-C<sub>30</sub>), branched fatty acids with iso- and anteiso- (n-C<sub>13</sub> to n-C<sub>17</sub>) configurations, n-alkanols (n-C<sub>12</sub> to n-C<sub>28</sub>), ketones, and sterols (Fig. 2). The distributions of n-alkanes, fatty acids and n-alkanols were dominated by high-molecular weight homologues, with ACL<sub>27-33</sub> of n-alkanes, ACL<sub>22-32</sub> of n-fatty acids and ACL<sub>22-28</sub> of n-alkanols averaging 28.7± 0.5, 24.0 ± 0.3 and 23.3 ±0.4 respectively, and showed little variation in the core (Fig. 2e). We also identified plant sterols (i.e. C<sub>29</sub> sterols

most abundant), pristane and phytane along with hopanoids (Fig. 3). The GDGT analysis showed that the distribution of all samples was dominated by brGDGTs. All samples had a BIT index of 1.0, and MBT'5<sub>ME</sub> index ranged between 0.24 and 0.48 (Supplementary Data). In general, the homogenous distribution of these biomarkers in the core are indicative of a terrestrial environment with higher plant components such as high molecular weight *n*-alkanes, *n*-fatty acids and *n*-alkanels and plant sterols.

### 4.2 Group 1: dry permafrost (0-35 cm)

The dry permafrost had the highest TOC and TN concentrations, averaging  $6.0 \pm 0.3$  and  $0.53 \pm 0.22$  mg g<sup>-1</sup> dry soil, respectively (Fig. 1c). The C:N ratio averaged of  $10.6 \pm 2.5$ , which is near the Redfield ratio and suggests balanced microbial growth (Fig. 1e & SFig. 1). The  $\delta^{13}C_{org}$  values were highest in the core, ranging from -22.1 to -20.0 %VPDB (Fig. 1d). The sum of odd *n*-alkane concentrations (avg. 628.2 µg g<sup>-1</sup> TOC; Fig. 2a), even *n*-fatty acids concentrations (avg. 1.67 mg g<sup>-1</sup> TOC; Fig. 2b) and even *n*-alkanols concentrations (avg. 432.1 µg g<sup>-1</sup> TOC; Fig. 2c) were all lowest in Unit 1. Within Unit 1, the CPI<sub>25-33</sub> of *n*-alkanes ranged from 3.7 to 6.9 (avg.  $5.5 \pm 1.2$ ; Fig. 2d) and the SC:LC ratio from 1.6 to 7.7 (Fig. 3b); both of which are the highest values found in the core. The highest CPI<sub>22-28</sub> of *n*-fatty acids were also found in Unit 1 (Fig. 2d), along with the ratio of  $100*C_{15-17}$  iso-FA+anteiso-FA/n-FA (up to 12.6; Fig. 3c). The dominant n-alkane chain length was  $C_{18}$  (Fig. 2f). Hopanoids, which were detected in all measured fractions (as hopanes, hopanols and hopanoic acids), where highest in Unit 1 (avg.  $84.6 \pm 65.3$  µg g<sup>-1</sup> TOC; Fig. 3a). Phytosterols such as stigmasterol and sitosterol and their derivatives were identified in highest concentrations in Unit 1, with the exception of the surface sample (Fig. 3g). Pr:Ph ratios were also highest in Unit 1 (avg.  $6.2 \pm 1.4$ ; Fig. 3e) and  $C_{30}$   $\alpha\beta/(\alpha\beta+\beta\alpha+\beta\beta)$  was <0.5 in unit 1 (Fig. 3d).

A sample taken at 10 cm below the surface was  $^{14}$ C dated and analyzed to determine the thermally partitioned splits. The sample taken at 10 cm below the surface yielded a bulk radiocarbon age of 41,749  $\pm$  1,736  $^{14}$ C yrs BP (F<sup>14</sup>C of 0.0055  $\pm$  0.0012; Table 1). The lowest temperature split (105 to 259°C) is composed primarily of small, mostly undiagnostic molecules that are mainly degradation products from larger (macro-)molecules (e.g. alkylbenzenes and other aromatics; Ginnane et al., 2024). In contrast, furans, which are found mostly in the 259 to 395°C partition (split 2) and 395 to 449°C partition (split 3), are diagnostic source indicators which can originate from carbohydrates, such as cellulose associated with the decay of terrestrial plants (e.g. Kaal, 2019). Pyrroles that are most abundant in split 2 are typical products of labile tetrapyrrole pigments (e.g. chlorophylls and its transformation products), whereas those found in split 4 are rather derived from the decomposition of plant lignin (Keely, 2006). The highest two temperature splits (splits 4 and 5 collected from 449 to 535°C and 535 to 750°C respectively) are comprised of compounds that are mostly derived from alteration of more complex macromolecules commonly associated with more relict carbon or from secondary thermal reactions during pyrolysis at higher temperatures (Ginnane et al., 2024). The resulting compounds derived in these fractions commonly consists of more stable, older material including kerogen, that is less accessible to bacterial degradation. Overall, the TOC-TN and biomarker indices suggest a higher level of organic carbon degradation by microbes relative to the underlying icy permafrost.

#### 4.3 Groups 2 and 3: icy permafrost (38-140cm)

The TOC and TN in the icy permafrost, averaging  $2.0 \pm 0.1$  and  $0.11 \pm 0.09$  mg g<sup>-1</sup> dry soil, respectively, were statistically lower than in the dry permafrost. The C:N ratios (16.4 to 53.7) were well above the Redfield ratio (Fig. 1e) The large range of C:N values in the icy permafrost may also partly be attributed to the low concentrations of N nearing the limit of analytical detection. The  $\delta^{13}C_{org}$  values showed a general decrease with depth: -21.4 ± 1.1 %VPDB in unit 2, and -25.4 ± 1.0 %VPDB in unit 3; (Fig. 1d). The sum of odd *n*-alkane concentrations (Fig. 2a), even *n*-fatty acids concentrations (Fig. 2b) and even *n*-alkanols concentrations (Fig. 2c) increased down core and were highest in Unit 3. The CPI<sub>25-33</sub> of *n*-alkanes ranged from 0.4 to 7.8 (avg.  $3.9 \pm 1.7$ ; Fig. 2d) and the SC:LC ratio, from 0.2 to 3.3 (Fig. 3b) within the icy permafrost, with little difference between unit 2 and 3. The dominant *n*-alkane was C<sub>18</sub> in Unit 2 (except for a dominance of C<sub>27</sub> at the ice table), whereas Unit 3 was dominated by C<sub>25</sub> (Fig. 2f). The sum of the concentrations of hopanoids decreased within the icy permafrost, averaging 59.5 ± 58.9  $\mu$ g g<sup>-1</sup> TOC in Unit 2 and 25.9 ± 18.0  $\mu$ g g<sup>-1</sup> TOC in Unit 3 (Fig. 3a). Phytane (Pr), pristane (Ph) and phytol were also detected in highest concentrations in Unit 3 (Fig. 3f). Pr:Ph ratios ranged from 0.2 to 2.2 in the icy permafrost and decreased with depth (Fig. 3e).

Figure 2 General distribution of biomarkers in core FHDP2C. a. concentration per gram organic carbon of low-molecular weight odd *n*-alkanes (*n*-C<sub>15</sub> to *n*-C<sub>19</sub>), mid molecular weight odd *n*-alkanes (*n*-C<sub>21</sub> to *n*-C<sub>25</sub>) and high-molecular weight odd *n*-alkanes (*n*-C<sub>27</sub> to *n*-C<sub>33</sub>), b. concentration per gram organic carbon of even low-molecular weight FAME (*n*-C<sub>14</sub> to *n*-C<sub>18</sub>), even mid molecular weight FAME (*n*-C<sub>20</sub> to *n*-C<sub>24</sub>) and even high-molecular weight FAME (*n*-C<sub>26</sub> to *n*-C<sub>30</sub>), c. concentration per gram organic carbon of even low-molecular weight *n*-alkanols (*n*-C<sub>14</sub> to *n*-C<sub>18</sub>), even mid-molecular weight *n*-alkanols *n*-C<sub>20</sub> to *n*-C<sub>24</sub>, and even high-molecular weight *n*-alkanols (*n*-C<sub>26</sub> to *n*-C<sub>28</sub>), d. carbon preference index (CPI) of *n*-alkanes (*n*-C<sub>25</sub> to *n*-C<sub>33</sub>), FAME (*n*-C<sub>22</sub> to *n*-C<sub>28</sub>) and *n*-alkanols (*n*-C<sub>22</sub> to *n*-C<sub>28</sub>) and, f. dominant *n*-alkane, FAME and *n*-alkanol chain length.

Figure 3 Diagnostic biomarker indices to differentiate between bacteria-derived and plant-derived organic matter. a. hopanes, hopanoic acids and hopanols, b. SC/LC (short-chain/long-chain) ratio of n-alkanes, c. ratio of iso-+anteiso- to straight chain FAME (n-C<sub>15</sub> to n-C<sub>17</sub>), d. C<sub>30</sub>  $\alpha\beta/(\alpha\beta+\beta\alpha+\beta\beta)$  hopane index, e. pristane/phytane ratio (Pr:Ph ratio), f. phytol, pristane and phytane and g. plant sterols.

Figure 4 Py-GC MS split composition for sample 2-C0-3, situated at 10 cm depth in the dry permafrost layer. a. compounds are classified approximately on bacterial- (B) or plant- (P) derived sources based on carbon number of alkanes and alkenes detected. b. RPO thermograph of CO<sub>2</sub> evolution and RPO and bulk radiocarbon measurements. Note: data points in red are indistinguishable from background measurements.

#### 5 Discussion

#### 5.1 Content and source of organic carbon: the legacy of the mid-Miocene tundra ecosystem

Soil organic carbon density in the dry and icy permafrost 140 cm core from Friis hills is about 4 times higher than those found in other ultraxerous soils like University Valley (SFig. 2; Faucher et al., 2017) and soils in the Mackay Glacier region (Van Goethem et al., 2020), but multiple orders of magnitude less than carbon stocks in mineral soils of maritime Antarctica (e.g. Alekseev and Abakumov, 2024; Simas et al., 2007). Currently, no vascular plants grow in the MDV (e.g. Virginia and Wall, 1999). As such, the low SOCd in most Quaternary-age soils in the MDV is attributed to the soil organic carbon being sourced either from endolithic micro-organisms (Faucher et al., 2017), glacially eroded material from older Cenozoic sediment and/or the Beacon Sandstone (e.g. Matsumoto et al., 1990a, Matsumoto et al., 2010), or Holocene age legacy carbon from aquatic systems (e.g. Lancaster, 2002, Barrett et al., 2006).

Conversely, the higher SOCd at Friis Hills is attributed to the tundra ecosystem that occupied the site during the mid-Miocene. The tundra macrofossil assemblage from the site included lichens, liverworts, mosses, dicots, grasses and sedges, along with

cuticles belonging to the *Nothofagaceae* family (Chorley et al., 2023). The lipid biomarkers that had a homogenous distribution in the 140 cm core are also consistent with the macrofossils assemblage. The BIT index of 1.0 derived from the brGDGTs samples suggests a fully terrestrial environment. The presence of high molecular weight *n*-alkanes, *n*-fatty acids and *n*-alkanols, along with phytosterols (i.e. sitosterol, stigmasterol and their respective stanols) and phytol (and pristane/phytane) are all indicative of a terrestrial environment dominated by higher order plants (Kögel-Knabner and Amelung, 2014; and references therein). The presence of mid-molecular weight (C<sub>21</sub>-C<sub>25</sub>) *n*-alkanes, *n*-fatty acids and *n*-alkanols indicate contributions of mosses and lichens. Thus, the Friis Hills sediments harbour a tundra-type biomarker signature and explains why Friis Hills has higher SOCd than other sites in the MDV (Fig. S3).

## 75 5.2 Degradation of organic carbon in the dry permafrost

390

395

The hierarchal clustering analysis of the samples in the 1.4 m core produced three groups and the variables and indices that were statistically different between the units. The dry permafrost (0-35cm) experienced a relatively higher degree of organic carbon degradation by microbes, whereas the underlying icy permafrost (38-71cm) experienced relatively lower levels of degradation.

The dry permafrost had C:N ratios near the Redfield ratio (Fig. 1e), higher δ<sup>13</sup>C<sub>org</sub> (Fig. 1d), a higher proportion of iso- and anteiso-FAs relative to *n*-FAs (Fig. 3c), low phytol (Fig. 3f), high Pr:Ph (Fig. 3e), low C<sub>30</sub> αβ/(αβ+βα+ββ) hopane index (Fig. 3d), and a higher contributions of low-molecular weight homologues of *n*-alkanes (Fig. 3b). All of these indicators can be attributed to a balanced microbial activity that degraded the ancient organic carbon in Unit 1. The relative higher concentration of phytosterols, pristane, phytane and phytol in the dry permafrost (Figs 3f & g), the dominance of hopanoids (Fig. 3a) and the absence of steranes, compounds that possess similar long-term preservation potential (Love and Zumberge, 2021), can all be attributed to the higher degradation of the plant material (Fig. 5). Moreover, the Py-GC-MS measurements in the thermally partitioned splits for a sample in the dry permafrost indicate a high level of refractory organic matter and degraded organic carbon.

Active microbial growth in dry permafrost has also been reported in other ultraxerous soils. For example, in University Valley, the soils that experience temperature above 0°C and where the icy permafrost is recharged by snowmelt, the C:N ratios were also distributed along the Redfield ratio (Fig. S2; Faucher et al., 2017). Similar ratios were found in the Mackay Glacier region of East Antarctica, supporting active nutrient cycling, although low respiration rates were associated with dormancy (Van Goethem et al., 2020). At Friis Hills, the ice table and underlying icy permafrost is being recharged by evaporated snowmelt (Verret et al., 2022). As such the dry permafrost transiently receives input of liquid water that could support microbial activity, similar to the endoliths growing in bedrocks (e.g. Friedmann, 1982), and to the water tracks that form in spring and summer (Chan-Yam et al., 2019). <sup>14</sup>C of bulk sample indicate active microbial activity is small (<1%; using a simple two-component mixing model and assuming F<sup>14</sup>C=0.5 for Holocene carbon, with a half-life of 5,730 yrs), but yet it is producing degradation of organic carbon over large time-scales (as shown in the biomarker results). As such, microbial activity is occurring at very

low rates, only when sufficient moisture is present in the dry permafrost layer. The rest of the time, the soil ecosystem appears to remain mostly dormant (e.g. Van Goethem et al., 2020).

## 5.3 Degradation of organic carbon in the icy permafrost







The degradation of organic carbon is not limited to the dry permafrost and it can also be observed in the underlying icy permafrost (unit 2: 38-71cm). Units 1 and 2 both display a higher concentration of hopanoids and higher SC:LC ratios than Unit 3 (Fig. 5). Additionally,  $\delta^{13}C_{org}$  values show a sharp shift below the ice table, and between Unit 2 and 3, where the average  $\delta^{13}C_{org}$  changes from -25.2 ± 0.7% to -21.4 ± 1.4% VPDB (Fig. 1d). This shift in  $\delta^{13}C_{org}$  values suggests preferential loss of  $^{12}C$  in CO<sub>2</sub> from respiration and organic matter degradation, which leads to sediment enriched in  $\delta^{13}C_{org}$ . The samples just below the ice table also have the lowest CPI<sub>n-alkane</sub> and CPI<sub>FAME</sub> (Fig. 2d), showing a higher degree of degradation.

These findings imply that rates of microbial organic matter degradation dominated over primary production of organic matter at a point in time following the mid-Miocene climate transition, which could not be precisely dated. The original biomarker signatures would have been preserved throughout the sediment column if the upper part of the permafrost column had remained permanently frozen until present day, preventing extensive bacterial degradation. However, recognizing the increased degradation state of plant-derived organic matter and higher contributions from bacterial lipids in the shallower part of the record suggests overprint of the original lipid distribution during more recent periods when the active layer thawed to that depth. Based on a study on <sup>10</sup>Be<sub>met</sub> concentrations in the upper section of the FHDP2C core, the onset of hyper-arid conditions occurred around 6.0 Ma (Verret et al., 2023); although it is uncertain if continuous or intermittent wet conditions prevailed throughout the late Miocene period and ended at 6.0 Ma. Either way, before 6.0 Ma, an active layer that seasonally thawed was present in at the Friis Hills, which could explain the higher degradability of Unit 2 compared to Unit 3. The transition from Unit 2 to 3 at ~80 cm depth could therefore represent the position of a paleo layer of increased biological activity within a relict active layer. Similar trends have been observed the in the Arctic, where paleo-active layers are identified biogeochemically by a higher degree of degradation. (e.g. Lacelle et al., 2019). Furthermore, permafrost carbon reservoirs have also been destabilized at a large scale in the Arctic during past warm periods of the Pleistocene-Holocene (e.g. Tesi et al., 2016). We show here that ancient permafrost carbon stocks in Antarctica, although marginal, could have also been destabilized during past warm periods in the near surface.

Figure 5 Boxplots of a. C:N ratio, b.  $\delta^{13}$ C, c. organic carbon, expressed in wt%, d. short-chain/long-chain *n*-alkane ratio, e. pristane/phytane ratio, f. ratio of iso-+anteiso- to straight chain FAME (n-C<sub>15</sub> to n-C<sub>17</sub>), g. C<sub>30</sub>  $\alpha\beta/(\alpha\beta+\beta\alpha+\beta\beta)$  hopane index and h-i. brGDGT ground temperature reconstructions for different units in core FHDP2C. These represent threshold temperatures for unlocking organic matter at different depths in the core: Unit 1 (0-35 cm depth), Unit 2 (38 -71 cm depth) and Unit 3 (93-140 cm depth). Units are based on hierarchical clustering.

#### 5.4 Temperature thresholds to reactivate biological activity in the dry permafrost and paleo-active layer




brGDGTs have been increasingly used to reconstruct past temperatures in Arctic permafrost because the methylation and cyclisation of brGDGTs can be correlated to ground surface temperature (e.g. Raberg et al., 2024). Previous studies in permafrost regions have made the assumption that microbial communities stored in permafrost reflect the environmental conditions at time of enclosement, not the current conditions (closed-system assumption; Kusch et al., 2019). However, this assumption does not take into account subsequent warm periods (which would result in a re-opening of the system). Therefore, the temperature reconstructions could correspond to either: (1) the temperature at time of enclosement (here the mid-Miocene) or (2) the temperature at time when the active layer last thawed to that depth and thus the threshold temperature to activate bacteria activity at a given depth. Since the previous sections have shown clear signs of overprinting in units 1 and 2, and to

some extent in unit 3, the latter assumption is more likely. Moreover, recent studies have shown that in cold regions, the brGDGT calibrations best represents the summer ground temperature, and not the mean air temperatures for which brGDGT results are typically calibrated (e.g. Raberg et al., 2024). However, in the ultraxerous zone of the MDV, the mean annual air temperatures approximates those at the ground surface temperatures since there is no vegetation, minimal snow cover and little organic material (Lacelle et al., 2016).







While soil calibrations for Antarctic sites are lacking, sites from Arctic permafrost in Svalbard, Greenland and Alaska are found in the global soil sample database of De Jonge et al. (2014) and these regions show the same dominance of pentamethylated and hexamethylated brGDGTs (Kusch et al., 2019). The BayMBT<sub>0</sub> calibration (Crampton-Flood et al., 2020) also offers a separate calibration that assumes brGDGT distributions only reflect months with mean air temperatures above freezing and was applied to the Antarctic Peninsula (Tibbett et al., 2022). We therefore used these two calibrations to reconstruct mean summer ground temperatures. Based on the conclusions from the previous sections, we attribute these temperatures in units 1 and 2 to be the threshold temperatures required to activate bacteria activity at different depths in core FHDP2C (November to February in the MDV; Obryk et al., 2020), while the minimal overprinting of biomarker signature in unit 3 likely represents conditions during the mid-Miocene. The distribution of all samples in core FHDP2C was dominated by brGDGTs. All samples had a BIT index of 1.0. MBT'5<sub>ME</sub> index ranged between 0.24 and 0.48 (Supplementary Data). We obtained temperatures varying between 2.3 and 7.1°C using the BayMBT<sub>0</sub> calibration (Crampton-Flood et al., 2020) and between -1.1 and 6.4°C using the De Jonge et al. (2014) calibration. The BayMBT<sub>0</sub> calibration yields slightly warmer (~1°C overall) temperature reconstruction than the DeJonge calibration (Figs 5h & i). The dry permafrost layer would require a mean summer soil temperature of ~2-4°C to reactivate biological processes. The temperature required to unlock the organic matter stored below the ice table seems a bit more uncertain but lies between  $0^{\circ}$ C near the ice table up to  $\sim$ 7°C down at 0.70 m (Figs 5h & i). The latter is in line with the mean summer air temperatures required to thaw the maximum active layer depth (based on meteoric Berylllium-10 concentrations) during the late Miocene (i.e. 7-10°C to thaw a maximum active layer depth of 2.74 m; Verret et al., 2023). In general, the temperature required to reactivate biological activity increases with depth since the thickness of the active layer is largely controlled by summer air temperature. The increase in threshold temperature to ~0°C near the ice table based on the De Jonge et al. (2014) calibration could be explained by the soil fauna having a high response rate to increased soil moisture (thawing ground ice) in the MDV (Niederberger et al., 2019; Andriuzzi et al., 2018). These temperature thresholds are in line with current conditions in the lower elevations of Taylor Valley at a site near Lake Fryxell  $(77^{\circ}36'06.1"\text{S}, 163^{\circ}08'19.6"\text{E}, 21 \text{ m a.s.l.};$  Bakermans et al., 2014), where the mean summer ground temperature is  $+2^{\circ}\text{C}$  and psychrophilic species have the ability to function. In. short, the organic matter in unit 3 has a low probability of being reactivated, while the section closest to the ice table, unit 2, has a much higher probability and hence is showing higher levels of degradation.

#### **6 Conclusions**

Although our study suggests that Holocene organic carbon is being introduced at high elevation sites such as the Friis Hills, modern C<sub>org</sub> contributions remain very low (<1%). Beyond the dry permafrost, C<sub>org</sub> is dominantly ancient and highly degraded. Based on the ramped pyrolysis approach to characterise and date different carbon pools, it seems like soil ecosystems in the high elevations (>1000 m a.s.l) of the MDV rely mostly on exogenous sources of C<sub>org</sub>, in this case legacy carbon from the Miocene tundra environment, but could have also marginally been able fix carbon in situ through the Holocene. More detailed 480 radiocarbon studies should be conducted to prove the latter. Moreover, based on our biomarker findings suggesting a gradient of C<sub>org</sub> degradation through the soil profile, we conclude that legacy carbon locked-in at depth in the permafrost has been bioavailable under past warmer climate post-deposition. Seasonal thawing during warm periods is at the origin of Corg degradation at depth. Such periods seem to be marginal over the last 14 Myrs. Carbon in the dry upper 35 cm of the core could be bioavailable at a mean ground summer temperature of  $+2^{\circ}$ C, conditions similar to those found in the lower elevations of 485 Taylor Valley. Carbon is a key physicochemical factor in the development of soil microbial communities (e.g. Cary et al., 2010). Therefore, future climate warming could lead to unlocking legacy sources of carbon which would cause considerable impacts on the structure and function of ecosystems in the MDV. Overall, the organic matter in the core appears to be compatible with a highly degraded signature of the mid-Miocene paleoenvironment, but also displays a gradual environmental over-print attributable to post-depositional conditions that is more important near the surface. This finding suggests that the 490 organic matter enclosed within the permafrost at Friis Hills is stable and bacterial alteration is mostly inhibited. However, bacterial degradation has occurred during warmer periods: episodically down to 80 cm and potentially beyond 140 cm during the late Miocene and down to 35 cm through to Holocene.

## Data availability

495 The data is uploaded as supplementary material.

#### **Author Contributions**

M.V., S.N., D.L., W.D. and K.N. designed this project and contributed to data analysis/interpretation and writing the manuscript. M.V. and S.N. undertook the lipid biomarker analyses, C.G. and J.T. provided the ramped pyrolysis oxidation and radiocarbon analyses, whereas S.N. analysed and interpreted the Py-GC-MS data. R.L. developed the Friis Hills Drilling Project. All authors edited the whole manuscript.

## **Competing interests**

500

One of the co-authors of this manuscript is a member of the editorial board of Biogeosciences.

## Acknowledgements

This work was funded by the New Zealand Antarctic Research Institute through an Early Career Researcher Seed Grant awarded to M. Verret in 2021. The Friis Hills Drilling Project (austral summer 2016–2017) was funded by the New Zealand Ministry of Business, Innovation and Employment through the Past Antarctic Science Programme (C05X1001) and the Antarctic Science Platform (ANTA1801) to R. Levy and T. Naish. We acknowledge Strategic Science Investment Funding (SSIF) from the New Zealand Ministry of Business, Innovation and Employment as part of the Global Change Through Time Programme (contract C05X1702). We thank A. Pyne, R. Pyne, H. Chorley and Webster Drilling for retrieving the cores at Friis Hills. A special thank you to N. Bertler for allowing us to use the GNS Ice Core Facility to store and sample the permafrost cores. Laboratory work was made possible with the help of the technical staff at the Sedimentology and Water Quality Laboratory at the Victoria University of Wellington (J. Chewings), the Geography Laboratory at the University of Ottawa (J. Bjornson), the Ján Veizer Stable Isotope Laboratory (P. Middlestead, W. Abdi and P. Wickham) and the Rafter Radiocarbon Laboratory, GNS Science New Zealand (J. Dahl and T. Ferrick).

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

Table 1: Summary of RPO-AMS results (Py-GC-MS chromatograms for the different splits presented in Fig. S4)

| NZA<br># | RP<br>Split | RP split<br>min<br>temp.<br>(°C) | RP split<br>max<br>temp.<br>(°C) | Split<br>size<br>(mg C) | CRA (yrs<br>BP) | CRA<br>Error | F <sup>14</sup> C | F <sup>14</sup> C<br>Error | Comments                                                          |
|----------|-------------|----------------------------------|----------------------------------|-------------------------|-----------------|--------------|-------------------|----------------------------|-------------------------------------------------------------------|
| 70400    | Bulk        |                                  |                                  | 1.07                    | 41,749          | 1,736        | 0.0055            | 0.0012                     |                                                                   |
| 75354    | 1           | 105                              | 259                              | 0.08                    | 42,935          | 26,742       | 0.0048            | 0.0159                     | This sample is indistinguishable from the background measurements |
| 75365    | 2           | 259                              | 395                              | 0.45                    | 36,576          | 2,379        | 0.0105            | 0.0031                     |                                                                   |
| 75361    | 3           | 395                              | 449                              | 0.22                    | 34,098          | 3,018        | 0.0143            | 0.0054                     | This sample is indistinguishable from the background measurements |
| 75364    | 4           | 449                              | 535                              | 0.32                    | 29,004          | 1,158        | 0.0270            | 0.0039                     |                                                                   |
| 75359    | 5           | 535                              | 750                              | 0.17                    | 31,716          | 3,971        | 0.0193            | 0.0095                     | This sample is indistinguishable from the background measurements |