# Peer review of "Preservation and degradation of ancient organic matter in mid-Miocene Antarctic permafrost"

_EGUsphere, 2025_

## Author Response (AR1)

**Editor comments**

I have evaluated the comments of two reviewers and your initial reply. I have also read the manuscript and am adding my comments below. Your reply entails that you agree to modify the text based on the reviewer comments in the revised manuscript. I would ask from my side a more bold restructuring of the figures, to aid the reader in understanding your interpretation of the biomarkers as environmental or degradation parameters.

We would like to thank to editor Cindy De Jonge for taking the time to read our manuscript and provide additional comments. We have restructured the figures for ease of understanding and address the remaining comments below.

**AE comments:**

Include key information on the age model that is shared in the review, in the manuscript text. The current information provided is not extensive enough. If the age model is done in comparison with a parallel core, add a figure with this comparison to the supp material and refer to it in the text.

We have added the information included in the reply to reviewers in the manuscript. Since the age model is discussed in great length in Chorley et al., 2023, we only provide what is relevant to the understanding of the presented work in this manuscript. The cores FHDP2A and FHDP2C are from the same location and correlate directly. Therefore, the information presented in Fig 11 of Chorley et al., 2023 applies to both core 2A and 2C. The section now reads as follows:

"The detailed age model of the FHDP cores is presented in Chorley et al. (2023). The lower section of core FHDP2C, which directly correlates with core FHDP2A, is constrained by 40Ar/39Ar dating of a tephra at 5.13-5.18 m depth (14.4 Ma; Fig 2). The most parsimonious solution for the normal chronology above the ash is that it is C5AD (~14.6 to 14.2 Ma), corresponding to the mid-Miocene climate transition (MMCT; Flower and Kennett, 1994; Shevenell et al., 2004). Furthermore, the analyzed sediments are from the upper part of sequence 13 from the Friis Hills composite presented in (Chorley et al., 2023). This sequence was likely deposited during a single precession cycle between 14.36 and 14.38 Ma (see Fig. 11 in Chorley et al., 2023) although we acknowledge that it may have been deposited during alternate precession cycles between ~14.4 and 14.2 Ma. Independent cosmogenic surface exposure dating at Friis Hills also suggests a minimum age of the surface of 11 Ma (Valletta et al., 2015)."

L 113. For future applications I would recommend to use a blank during the drying stage as well, as direct contact between the samples and PE bags can liberate organic compounds. Using a few blanks to quantify this process, will allow you to correct for this.

Thank you for the recommendation. We are aware that the use of blanks is preferable and will take this recommendation for future applications. The drying of the samples was done before the research team considered doing biomarker analysis, in order to comply with regulations from the importing authorities.

Your reply entails that the GDGT distribution can be interpreted in two ways, I agree that the downcore interpretation of this record can be a mixture of both initial soil formation signal and a signal produced during later degradation. Please carry on this dual interpretation of the GDGT distribution until the final conclusion of the ms.

Units 1 and 2 likely represent overprinting, while GDGTs likely reflect conditions from the mid-Miocene (based on the biomarker results). We added a few details to justify the preferred interpretation. The beginning of section 5.4 now reads:

"brGDGTs have been increasingly used to reconstruct past temperatures in Arctic permafrost because the methylation and cyclisation of brGDGTs can be correlated to ground surface temperature (e.g. Raberg et al., 2024). Previous studies in permafrost regions have made the assumption that microbial communities stored in permafrost reflect the environmental conditions at time of enclosement, not the current conditions (closed-system assumption; Kusch et al., 2019). However, this assumption does not take into account subsequent warm periods (which would result in a re-opening of the system). Therefore, the temperature reconstructions could correspond to either: (1) the temperature at time of enclosement (here the mid-Miocene) or (2) the temperature at time when the active layer last thawed to that depth and thus the threshold temperature to activate bacteria activity at a given depth. Since the previous sections have shown clear signs of overprinting in units 1 and 2, and to some extent in unit 3, the latter assumption is more likely."

We add a clarification on this interpretation later in text:

"Based on the conclusions from the previous sections, we attribute these temperatures in units 1 and 2 to be the threshold temperatures required to activate bacteria activity at different depths in core FHDP2C (November to February in the MDV; Obryk et al., 2020), while the minor overprinting of biomarker signature in unit 3 likely represents conditions during the mid-Miocene."

L 16. If the interpretation of the 14C dating is not conclusive, I would rephrase this in your abstract ("we aim to constrain").

We have followed the recommendation from reviewer 1 and rephrased as such:

"Here, we explore the additions of modern organic carbon, and the preservation and degradation of organics and lipid biomarkers, in a 1.4 m mid-Miocene age (~14.5-14.3 Ma) permafrost soil column from Friis Hills."

For the n-alkane distribution description at L 170, the authors seem to suggest that all n-alkane plant waxes derive from fatty acids. I don't agree with this statement, please have a look at the paper here (https://doi.org/10.1111/nph.16571) for clarification. If you disagree and have different reference for this, please add it to your manuscript text.

The text in section 3.4.1. now reads:

"The carbon preference index (CPI) is a parameter that quantifies the ratio of odd to even n-alkanes and ratio of even to odd n-fatty acids based on its carbon number and is an indicator of the degradability of organic carbon. Natural distributions or well-preserved n-alkane signatures are expected to show a predominance of odd-numbered n-alkanes and even-numbered n-fatty acids corresponding to high CPI values. The CPI value decreases with ongoing alteration of organic carbon (Bray and Evans, 1961):"

Combine Fig 1&2, potentially place Fig. 1B (ice table depth) in the supplementary.

We have combined Figs 1 & 2 and changed reference to figure in text accordingly. We have added Fig b to Supplementary Figures.

I agree with the comment of reviewer two that the amount of figures is currently too high. In my opinion, this makes the manuscript much more difficult to follow, and limits the impact of your study. Currently, the reader has to jump too often between figures. My suggestion: Based on your clustering, the interpretation of proxies can be done in groups of environmental variability, instead of per compound class (original compound class figures can be included in the supplementary). Combine key parameters of Fig. 3, 4, 5 and 6 in 1 (or max 2) overview figure. Consider grouping the proxies per parameter or based on the clustering result (fi. bacterial biomass, plant biomass, climate, ...).

We have reduced the number of figures by grouping proxies. We have removed the HPFA overall from text and figures to lighten the text and reduce the number of indices.

Figure 2 now consists of more general biomarker proxies and Figure 3 of more diagnostic indices to differentiate plant-derived and bacteria-derived organic matter. We have changed the references in text accordingly.

Fig. 8 and 9 could also be included in an overview figure, these summary statistics can be included as a side panel to the downcore plot of the these ratio values. Or at the least, they can be combined in a single figure.

Figures 8 and 9 have been grouped together. The downcore figure already having numerous panels, we decided to present the boxplots separately.

L 426: These calibrations are based on the same ratio, and will thus always show the same trends. Their agreement thus not imply that the temperature reconstruction is more realistic or accurate.

We removed the statement about similar trends to avoid suggesting that their agreement is more accurate.

**Reviewer 1**

Verret et al. reports microbial organic matter degradation of a mid-Miocene age permafrost soil column from the McMurdo Dry Valleys (MDV), especially near the surface in the 'dry permafrost', which likely occurred during warm periods post-deposition. In addition, the lower 'icy permafrost' presents evidence of higher plants from a tundra ecosystem analogous to present-day southern Greenland. In order to study the degradation/preservation of organic matter, the author utilises bulk sediment TOC and  $\delta^{13}$ C, and biomarker distributions. A hierarchical clustering of this dataset highlights three distinct depth intervals that are attributed to increasingly lower levels of organic carbon degradation downcore. Moreover, the author uses two calibrations (i.e. MBT'5ME and BayMBT0) to acquire temperature reconstructions from brGDGTs. Both present a similar range of values and provide insight on the warming required to "unlock" the carbon stored in the permafrost for microbial activity.

Overall, this paper is articulately written with only a few incorrect spelling and/or grammar mistakes. The figures are also very clear and easy to interpret. The Introduction (section 1) and the Study Site (section 2) provide a succinct overview of the locality (i.e. MDV) and the rationale behind the research undertaken. It is nice to see a study that takes advantage of a multi-lipid approach, and not just the compounds most commonly applied (such as n-alkanes and fatty acids). The data also provides insight into high-latitude systems, which are considerably under-studied. The potential impact of carbon released from permafrost, as a response to future warming, is still a major gap in our knowledge, and fingerprinting the biogeochemistry of Antarctic soils would be a crucial step towards developing a better understanding.

Although written eloquently, I believe that the readers would benefit from the author providing geographical context in the introduction (see mostly minor comments for suggestions on improvements). In addition, please find suggestions for how the abstract (see Section 4) and results (see Section 3) could be developed slightly. Finally, there are a few interpretations that I find myself struggling to follow, and believe would benefit from more clarification (see Section 1). In particular, there are major questions pertaining to use of the term "degradation" that a slight reframing of the discussion may solve (see Section 2.1), and potential further exploration of hopanoids (see Section 2.2).

We thank Emily for her detailed insight and useful feedback on our manuscript. It is clear from the comments that she has read our manuscript diligently. We address her comments in the following sections.

**Major comments:**

- 1) Interpreting the changes in biomarker distribution
- 1.1 A new source of organic carbon post-Miocene?

The author suggests that the overall biomarker distribution support a tundra ecosystem that was established during the mid-Miocene (section 5.1). This includes greater values in the SOCd, BIT index, and abundance of high-molecular weight compounds, phytosterols, and phytol. Although the macrofossils further confirm a tundra ecosystem, how confident is the author that the deposits are only composed of mid-Miocene sediments? As described in section 2, the dating relies on magnetostratigraphy constrained by a tie point at much lower depths (tephra layer at 5.13-5.18 m). Is this at a resolution that provides robust ages for the 140 cm section studied? Although the Py-GCMS data provides evidence of plant material, could plants not have grown during warm intervals post-Miocene? Did a tundra ecosystem only exist during the Miocene or is the composition/species of macrofossils specific to the Miocene? If the latter, could the author cite some papers to support this? I ask for clarity on the age model as I wonder how the author ruled out the possibility that the distinct change in the biomarker distribution of unit 1 represent a different time interval with a shift in the depositional environment, and thus lithology. Especially considering the 14C analyses at ~10 cm yielded an age of ~42 kyrs (even in splits 4 and 5, that supposedly contain the "older material"; line 293), and the TOC is much higher in unit 1.

**Based on the editor's comments, we have reinforced this section (read section on editor comments)**

We also added a statement on line 140 to remind the reader of the statements detailed in lines 114-122.

**1.2 Microbial organic matter degradation during deposition in the Miocene?**

In addition to clarifying why the change in the biomarker distribution of unit 1 does not represent a change in source, and instead is a degradation signal, how did the author determine that microbial organic matter degradation was less severe in the underlying icy permafrost? Besides the abundance of plant-derived degradation products (e.g. pristine and phytane), most of the evidence in the dry permafrost is indicative of a contribution of bacteria-derived organic carbon (e.g. low C:N ratios, high  $\delta$  13Corg values, and high relative abundance of iso-FAs, LMW n-alkanes, and hopanoids). I think inferring

microbial organic matter degradation from multi-lipid evidence of microbial activity is not a bad assumption, however does this equate to better preservation of lipids in the icy permafrost? (see Section 2.1 for further discussion and a potential solution). It seems unlikely that microbial organic matter degradation did not take place when the permafrost soil was accumulating during the Miocene. Especially considering that there is evidence of higher plants, which implies warmer and wetter conditions that would also favour microbial activity (Chorley et al., 2023). Therefore, is it more nuanced to state that the dry permafrost is influenced by a faster rate of microbial organic matter degradation compared to primary productivity?

As explained in our response to the previous question, we can robustly infer that the organic matter in unit 1 is of Miocene age, derived from plants living during a warmer climate, and can exclude any contributions due to the absence of plant vegetation in later periods. We agree that active microbial degradation would have also occurred during the mid-Miocene, as suggested by the reviewer. Our results even suggest that the differences between units 2 and 3 could represent another paleo-active layer. We know from previous studies that there was an active layer with liquid water incorporating the sections of the sediment core through the late Miocene (Verret et al., 2023). We therefore interpret that the signature found in this section of the core would reflect the most recent active layer, altering the original Miocene signature. Therefore, our explanation regarding higher degradation near the top and lower near the bottom is derived from the fact that there is a gradient of less plant-derived lipids and more bacteria-derived lipid near the surface. Hence, there is a depth-dependent degradation trend where unit 1>unit 2>unit 2>unit 3.

As suggested by the reviewer, we added a nuance in text in line 537: "These findings imply that rates of microbial organic matter degradation dominated over primary production of organic matter at a point in time following the mid-Miocene climate transition, which could not be precisely dated. The original biomarker signatures would have been preserved throughout the sediment column if the upper part of the permafrost column had remained permanently frozen until present day, preventing extensive bacterial degradation. However, recognizing the increased degradation state of plant-derived organic matter and higher contributions from bacterial lipids in the shallower part of the record suggests overprint of the original lipid distribution during more recent periods when the active layer thawed to that depth."

- 2) Gradient of organic carbon degradation that decreases downcore
- 2.1 Defining "degradation"

Another reason for questioning the statement that there is lower levels of organic carbon degradation downcore, is that, although the Py-GC-MS data presents degraded organic matter in the dry permafrost, there is no Py-GC-MS data from the icy permafrost to compare this to. Potentially, it is actually comparatively much lower in concentration. Moreover, the CPI suggests degradation downcore. Typically, thermally mature organic matter exhibits low CPI values (~1), and high CPI values (>3-30) indicates relatively unaltered organic matter (Diefendorf & Freimuth, 2017). The dry permafrost coincides with CPI values >3. I think the solution here would be to differentiate between microbial degradation and abiotic degradation that has occurred over long timescales to the Miocene deposits. Reframing the paper to fit these definitions would massively help the confusion caused by what comes across as contradictory statements throughout. For example, "Beyond the dry pemafrost, Corg is dominantly ancient and highly degraded" (line 444) and "Overall, the organic matter in the core appears to be compatible with a highly degraded signature of the mid-Miocene paleoenvironment" (line 455). This will improve what appears to be a lack of consistency between the abstract and conclusion (see more recommendations in Section 4).

As part of this recommendation, I would like to suggest that the author focuses on reframing the following section in the Introduction (section 1; line 53):

"Low-molecular weight organic compounds (e.g. sugars and amino acids) and lipids with double bonds or polar functional groups (e.g. fatty acids and alcohols) are typically susceptible to microbial decomposition. However, other lipid biomarkers, such as apolar, saturated hydrocarbons (e.g. alkanes), isoprenoids (e.g. phytane) and cyclic compounds (i.e. hopanes or steranes), are refractory compounds formed during diagenesis (Peters et al., 2007) and may be preserved for long geological timescales (e.g. Eigenbrode, 2008)."

In this section, the definitions can be introduced. Also, as it currently reads, since there is a dominance of easily degradable compounds in the dry permafrost and refractory compounds in the icy permafrost, this suggests lower levels of organic carbon preservation downcore.

We don't want to overinterpret the biomarker data and split indices into "biotic" and "abiotic". Since our material is very old, we have long periods with good preservation, interrupted with intervals where we have biotic degradation which could be from long time ago. The bacterial degradation is not exclusively happening during modern times, because the majority of the organic matter at the surface is very old too. However, it should also be noted that the old age would also partially result from the high age of the material that gets degraded, so if bacteria decompose the ancient material, it will also include some of the very old signature into their biomass too, which makes it look older than it may be, so it is quite difficult to be sure when the overprint really happened.

We added a statement in the introduction to resolve the contradiction between abiotic and biotic degradation (line 73): "Over large time-scales, both biotic and abiotic degradation occurred at Friis Hills. Abiotic degradation would result in the organic material near the surface to be better preserved (due to the principle of superposition). However, in a permafrost environment, biotic degradation which is restricted to the active layer (layer that thaws in freezes on an annual basis) during warmer climate intervals would overprint the abiotic degradation and would be more important near the surface, incorporating both contemporary and ancient organic carbon into the system (Kusch et al., 2021) "

2.2 Exploring the hopanoids to determine abiotic degradation This is a recommendation that the author can chose to ignore, however the hopanoid dataset may provide interesting insight into abiotic degradation occurring at this site, further supporting the CPI results. I make this suggestion as the author has already identified and integrated the hopanoids, therefore it is information that would take little work to acquire. If hopanes and their various stereoisomers were discovered, there a multiple ratios that can be used for tracing thermal maturation. For example, the C-17 and C-21 position of hopanes change from a  $\beta\beta$  configuration to a more stable  $\beta\alpha$  configuration during early diagenesis (Farrimond et al., 1998).

Thank you for the recommendation. We had actually explored hopane maturation and now include the  $C_{30}$   $\alpha\beta/(\alpha\beta+\beta\alpha+\beta\beta)$  index in Fig. 6e. We introduce it in the method (line 257) section, and added it to the results (line 400) and the discussion (line 506).

**3) Developing the methods and results section**

The comment on the methods section is minor, but some of the indices which are eventually discussed in the results are not introduced in the methods. This includes the cholesterol to plant sterol ratio, 100\*C15-17 iso-FA+anteiso-FA/n-FA; Pr:Ph, and C26-28 to C22-24 n-alkanols. It would be useful for the readers to be able to refer back to the methods to understand the ratios and what they represent. Having said this, Pr:Ph (Figure 6 and 8) and C26-28 to C22-24 n-alkanols (Figure 5) are not actually discussed within the text anywhere...is it required in the figures? On the other hand, TN is discussed (line 269 and 298) but is not available in a figure.

To streamline the result section and keep the focus on important indices, we removed the cholesterol to plant sterol ratio and C26-28 to C22-22 n-alkanol ratios (in Figure 5).

Pr:Ph is discussed in text in sections 4.2, 4.3, but we added a statement about it in section 5.2.

We added a section "3.4.5. Other indices derived from saturated hydrocarbons, isoprenoids and fatty acids" to make sure that all the indices are discussed in the method section, where 100\*C15-17 iso-FA+anteiso-FA/n-FA, Pr:Ph and  $C_{30}$   $\alpha\beta/(\alpha\beta+\beta\alpha+\beta\beta)$  are discussed.

For the results section, although I agree with the author's choice to first discuss the general background composition of biomarkers, prior to describing the data within each unit in section 4.2 and 4.3, I feel the first sentence is somewhat misleading. The author lists the compounds that show little variation, however then goes on to discuss their differences. For example, n-alkanes are stated as a biomarker that does not vary much, yet the CPI (which compares the odd vs. even n-alkanes) and SC:LC ratio (which compares short- vs. long-chain n-alkanes) is highest in unit 1. As done so for ACL, the author could instead focus on the averages or ranges of values. In addition, I would advise the author to be careful with not interpreting the data too much at this point. The fact that the cholesterol to plant sterol ratio and the BIT index indicates a terrestrial setting and the high-molecular weight compounds further support higher plant input is useful, however to infer that this reflects a "tundra dominated ecosystem" and this agrees with interpretations made from "macrofossils" is something that could be saved for the discussion. Similarly, in section 4.2, I would remove the sentence starting with "Overall..." and maybe expand on what the biomarker indices described in the first paragraph indicate, as done so for HPFA on line 278. This also applies to section 4.3.

Thank you for picking up on this contradiction. We rephrased the first paragraph of section 4.1 to reframe the general trends and avoid over-interpretation. It now currently reads: "We identified the following components within the 1.4m core: n-alkanes (n-C15 to n-C34), isoprenoids, hopanoids, n-fatty acids (n-C12 to n-C30), branched fatty acids with iso- and anteiso- (n-C13 to n-C17) configurations, n-alkanols (n-C12 to n-C28), ketones, and sterols. The distributions of n-alkanes, fatty acids and n-alkanols were dominated by high-molecular weight homologues, with ACL27-33 of n-alkanes, ACL22-32 of n-fatty acids and ACL22-28 of n-alkanols averaging 28.7± 0.5, 24.0 ± 0.3 and 23.3 ±0.4 respectively, and showed little variation in the core (Figs 3d, 4d & 5d). We also identified plant sterols (i.e. C29 sterols most abundant), pristane and phytane along with hopanoids (Fig. 6). The GDGT analysis showed that the distribution of all samples was dominated by brGDGTs. All samples had a BIT index of 1.0, and MBT'5ME index ranged between 0.24 and 0.48 (Supplementary Data). In general, the homogenous distribution of these biomarkers in the core are indicative of a terrestrial environment with higher

plant components such as high molecular weight n-alkanes, n-fatty acids and n-alkanols and plant sterols."

We also removed the sentence in section 4.2: "Overall, these biomarkers reflect the background composition of the tundra-dominated ecosystem of the site during the mid-Miocene as inferred from the macrofossils."

We removed the sentences in section 4.3: "Overall, these indices display a gradient of organic carbon degradation that decreases downcore through the icy permafrost. The hierarchal clustering analysis suggest higher level of organic carbon degradation in the dry permafrost, intermediate level in unit 2 and lower level in unit 3."

**Following input from the editor, we have removed the HPFA altogether.**

**4) Consistency between the abstract and conclusion**

In addition to being more cautious with the use of the term "degradation" in the conclusion, to avoid what could be interpreted as contradictory statements between the abstract and conclusion (see Section 2.1), the abstract could also benefit from providing a more 'complete summary' of the study.

The abstract appears to focus on the loss of the "legacy" mid-Miocene age carbon in the upper 'dry permafrost', however does not delve into the source of carbon that overprints this signal, as done so in the conclusion on line 443, i.e "Although our study suggests that Holocene organic carbon is being introduced at high elevation sites such as the Friis Hills, modern Corg contributions remain very low (<1%)." This would be useful to include, in addition to a sentence on line 20 that explicitly states what the evidence found within the 'dry permafrost' indicates, as done so for the icy permafrost on line 21, i.e. 'the multiple proxies indicate contribution of bacteria-derived organic carbon, and thus microbial organic matter degradation.' Moreover, the discussion surrounding the brGDGT results and the significance of temperature on carbon availability / microbial activity is not mentioned in the abstract. However, although I feel the abstract could better expand on the breadth of data and interpretations made, I also acknowledge that there is a word limit. If the addition of the source of carbon and the temperature reconstructions is difficult, I recommend the author prioritises honing the final sentence of the abstract to make it more impactful (line 25). What exactly is the key outcome that is novel to this study that the author would like to highlight here? Is it really that this record provides a nice archive to examine ecological changes in the past? How about the cautionary tale of needing to exclude more recent microbial activity prior to interpreting biomarkers in sedimentary deposits? Or even that these archives are vulnerable to

future warming? Alternatively, what we can learn from "present-day organic processes" (line 63)?

We added a statement on line 21: The near-surface (upper 35 cm) dry permafrost has lower C:N ratios, higher  $\delta^{13}$ Corg values, higher proportion of branched fatty acids with an iso- and anteiso- configuration relative to n-fatty acids, lower phytol abundance and higher contributions of low-molecular weight homologues of n-alkanes, than the underlying icy permafrost, indicating higher contributions from bacteria-derived organic matter.

We added the following statement on line 26: "This implies that legacy mid-Miocene age carbon in the near-surface soils (c. 35 cm) has been prone to microbial organic matter degradation during times when the permafrost thawed, likely during relatively warm intervals through the late Neogene (~6.0 Ma) and sporadically during the Holocene (<1%), when ground summer temperatures were ≥+2°C (based on brGDGTs temperature reconstructions)."

We believe that the current last sentences convey our message that ancient organic matter is preserved through large time-scales at Friis Hills, but also vulnerable to past and future climate change. More details are discussed in the conclusion.

**Minor comments:**

Line 15: "low input rates" of what exactly?

We clarified the sentence, which reads now: "The occurrence of ancient soil organic carbon combined with low accumulation of contemporary material makes it challenging to differentiate between ancient and modern organic processes."

Line 15: replace "document" with explore or a verb that is more related to the aims of the study rather than results at this stage of the abstract

We changed "document" to "explore".

Line 17: define ages for mid-Miocene, e.g. '(xx-xx Ma)'

Added (~14.5-14.3 Ma).

Line 18: define FA, e.g. 'branched fatty acids with an iso- configuration relative to n-fatty acids' (following how the author defined iso-FAs in line 249).

Changed to "branched fatty acids with an iso- configuration relative to *n*-fatty acids".

Line 19: go through the whole manuscript to check whether a hyphen is used or not used between

"low/mid/high" and "molecular" and make it consistent

Added hyphen throughout text.

Line 24: define ages for late Neogene, e.g. '(xx-xx Ma)'

This is not derived from formal age range, but we added (~6.0 Ma).

Line 24: Since this study specifically focuses on lipids, "Biomolecules" could be replaced with 'lipid biomarkers'

Done.

Line 27: it may be useful to clarify to the readers that "The hyperarid polar desert" refers to the MDV e.g. 'The MDV is a hyperarid polar desert, and amongst the coldest and driest environments on Earth (e.g. Horowitz et al., 1972). The MD can be divided into...'

Changed the wording to "This hyperaid polar desert"

Line 40: specify where the "Shackleton Glacier region" is in relation to MDV? e.g. 'further inland'

Done.

Line 42: could refer to Fig. 1a here, to place University Valley in geographical context

Done.

Line 43: the "3" needs to be in subscript, e.g. "CaCO3"

"CaCO3" as a whole is already in subscript, so we are not able to use an additional level of putting this into subscript. Perhaps this can be done in editing.

Line 45: add r to 152 ky?

Done.

Line 46: Fig. 1a can again be referred to here, after "At Friis Hills..."

Done.

Line 47: the "oldest" what? permafrost?

Yes, we added permafrost again for clarification.

Line 48: define ages for mid-Miocene again in introduction, e.g. '(xx-xx Ma)'

Added (~20-14.6 Ma)

Line 56: could be reworded to, e.g. 'and may be preserved on geological timescales'

Done.

Line 59: start new paragraph?

In our version of the manuscript this already is a new paragraph.

Line 59: could remove "diagenetic"

Done.

Line 60: for consistency within the text, use total organic carbon (TOC) rather than "total Corg". In addition, technically speaking, total N is not part of "bulk organic carbon analysis".

Changed to TOC. Changed to "bulk carbon and nitrogen analyses"

Line 93: is there a map of Friis Hill with the three sites? Could be added as a supplementary figure?

These three "2" boreholes are few meters apart. A map wouldn't be useful here. Since we only discuss the 2C borehole here, we think the coordinate is suitable for situating the site.

Line 95: "metre" can be abbreviated

Done.

Line 117: this sentence could be reworded, e.g. 'The total organic carbon (TOC) and total nitrogen (TN) were measured to determine: i) the soil organic carbon density

(SOCd); and ii) if the C:N ratios in the bulk sediments follow the Redfield biological stoichiometry ratio (6:61, Redfield, 1934).'

**Done.**

Line 162: could a more specific word be used instead of "measurements"? e.g. 'The concentrations or abundances of lipid biomarkers were used...'

**Done.**

Line 171: this sentence is repeating what was stated in line 166, and I would also remove "as odd chains get altered into smaller chain lengths" as it may confuse readers considering CPI does not capture changes in chain length. If this sentence is removed, line 169 could be incorporated into the paragraph beginning on line 165.

Sentence line 171 was removed and sentence line 169 was incorporated into line 165.

Line 185: should "i" and "Ci" be in italics to match the Equation 3?

**Done.**

Line 195: although Strauss et al. (2015) uses the term "chemical degradation", could the author clarify what this means? Is this an abiotic process not involving microbial activity?

**Changed "chemical" to "organic matter".**

Line 198: could reword "Since n-alkanes are typically preserved better..." to, e.g. 'less labile' or 'more stable' etc.

**Changed to more stable.**

Line 198: is there a threshold or range for "low HPFA" values that represent "high degrees of organic carbon degradation." I also wonder if it would be safer to say "relatively high" because a "high degree" could be interpreted as catagenesis or even metagenesis of organic matter, unless this is what the author intended? This also applies to line 278 and 361

Following input from the editor, we have removed the HPFA altogether.

Line 201: could the author define the BIT index, e.g. 'branched/isoprenoid tetraether (BIT) index'

**Done.**

Line 201: is the equal sign most appropriate here? i.e. must the BIT value equal 1 to indicate a terrestrial signal and vice versa for 0 and a marine signal?

**Changed to ≈.**

Line 210: a space is missing between the subsection 3.5 and paragraph below

**Done.**

Line 215: RPO is used here and in the captions for Figure 7 and Table 1, without it being defined anywhere in the text

**Defined acronym.**

Line 223: define OM, e.g. 'organic matter'

Removed acronym all together from manuscript.

**Reviewer 2**

The manuscript by Verret and co-authors examines a Miocene age sediment sequence from the Friis Hills in Antarctica, to explore the potential lipid biomarker and organic carbon evidence for both the original Miocene tundra soils and the subsequent diagenetic impact on the composition of the organic matter. Several techniques and metrics are applied, including a characterisation of a wide range of lipid distributions, carbon and nitrogen content, bulk carbon isotopes, pyrolysis and radiocarbon analysis. The authors argue that they show evidence for the original tundra organic inputs, of Miocene age, and a later diagenetic impact during warmer periods of Antarctica's past, when the formation of the active layer would likely have enabled microbial activity. The study is an interesting one, which also contains a rich set of data which has been carefully produced and presented.

We would like to thank the reviewer for taking the time to read through our manuscript and provide valuable feedback.

I have two main concerns.

First, my main concern is about the framing of the temperature story. The authors argue that the soil temperatures that they reconstruct are indicative of "threshold temperatures" required to support microbial activity and thus organic matter degradation (e.g. line 63 and Section 5.4). They use the branched GDGT proxy to calculate the soil temperatures from each of their three sedimentary units. I am unclear why temperatures have to represent post depositional diagenesis, which the authors argue occurs perhaps millions of years after deposition, when branched GDGTs are found with a global distribution today including in the polar regions, and importantly including in tundra settings. Why are the brGDGTs not a contemporary signal of tundra soil temperature during the Miocene, with a potential cooling over time being represented moving upwards from unit 3 to 2 to 1? In line 253-254, for example, the text includes mention of the dominantly terrestrial origin of the brGDGTs in amongst text which refers to the lipid biomarkers being "...indicative of a terrestrial environment ". This is also how we would interpret brGDGTs in more recent sedimentary sequences: as a tool to explore temperature change in the context of co-recorded signals of environmental change. To me, a change in temperature over time (from 7°C deeper in the site to near zero near the top) is more easy to explain than how an activation temperature of 7°C is needed at depth if the surface is only recording 0°C? The authors seem to be suggesting that once the soil hits a single temperature, the microbes record only that temperature, even if it gets warmer than that afterwards?

The climatic transition occurred several million years ago and our dataset seems to point towards a subsequent alteration of the organic material post-deposition, at least in units 1 and 2. This is why we present the temperature reconstructions as threshold temperatures for microbial activity. Had there been no sign of biological overprinting near the surface, the Miocene climatic transition signature recorded in the brGDGT that the reviewer suggest here would have been preferred. However, since the organic material has been altered post-deposition, the brGDGTs would reflect overprinting. In a permafrost context, biological activity is restricted to the active layer that thaws and freeze on an annual basis. The thaw depth is temperature dependant. Therefore, the 7°C required to reactivate activity at depth is simply a function of the surface temperature required to induce a thaw depth of 1m at this site.

We added a statement on line 580 to clarify this: "However, this assumption does not take into account subsequent warm periods (which would result in a re-opening of the system). Therefore, the temperature reconstructions could correspond to either: (1) the temperature at time of enclosement (here the mid-Miocene) or (2) the temperature at

time when the active layer last thawed to that depth and thus the threshold temperature to activate bacteria activity at a given depth. Since the previous sections have shown clear signs of overprinting in units 1 and 2, and to some extent in unit 3, the latter assumption is more likely."

**Based on the editor's comments, we have reinforced this section (see section on editor comments)**

Second, radiocarbon analysis seems under explored. The authors argue that the fraction modern carbon suggests less than 1% input of microbially produce carbon, but how this figure is arrived is unclear, when they only refer to the bulk date (line 378) and do not explore the range of values given by the RPO in figure 7. If microbial activity is using both Miocene (infinite age) and modern (atmospheric) carbon then I would expect a large imprint of the infinite age carbon on this signal, which might mean an underestimate of microbial activity. I think detail is needed here to explain how the authors reach this <1% figure.

We don't want to overinterpret the radiocarbon data since there is considerable uncertainty of what is considered "modern". It is important to note that modern biological processes would also incorporate ancient carbon (Kusch et al., 2021). Our aim here was to detect any sort of radiocarbon active contribution. The <1% modern input relates back to a simple two-component mixing model and for  $F^{14}C=0.5$  (Holocene) and  $F^{14}C=0$  (ancient) and using the bulk date ( $F^{14}C=0.0055$ ). Younger ages yielded from the RPO would include a larger fraction of Holocene carbon, although the selection of  $F^{14}C$  value for this "modern" carbon makes quite a difference.

We added a specification on this on line 504:  $^{14}$ C of bulk sample indicate active microbial activity is small (<1%; using a simple two-component mixing model and assuming  $F^{14}$ C=0.5 for Holocene carbon, with a half-life of 5730 yrs), but yet it is producing degradation of organic carbon over large time-scales (as shown in the biomarker results).

**General comments:**

• it's unclear why only one sample near the surface is analysed for pyrolysis, as repeating these measurements in units 2 and 3 might have shown what a less degraded horizon looked like

With limited resources, we targeted the unit most likely to have modern contributions (largest overprinting), making a sample from unit 1 the most important to run. We agree with the reviewer that running the pyrolysis experiment on one sample from each unit would have been optimal, but we only had the resources to analyse one sample (6 radiocarbon dates in total). However, we wanted the paper to focus mostly on the biomarker distribution, since the timing of activity is difficult to determine. We ran the pyrolysis analysis in parallel, so we thought it was still interesting to present the results here. We acknowledge in the conclusion that the dating should be interpretated with caution and recommend that it should be investigated further.

 I suggest that the authors think carefully about which graphs to include or on which order. Figure 2 is presented within the text early on, but the results section lists figures 3-5 before then, and the latter graphics contain a lot of detail which is confusing because the opening section says many of the lipids don't show any change, so I'm unclear why all of these results are displayed in the main text.

We think it is important to showcase Figure 2 first as it presents the stratigraphic units in relation to the 3 units presented in this paper. We added a reference to it earlier in the results on line 322.

**Based on the editor's comments, we have made significant changes to the figures (see section on editor comments)**

**Minor clarifications:**

• line 113: there is a risk of organic matter degradation if samples are warmed up, so I'm not clear why the authors would not freeze dry these samples rather than exposing them to water and air at elevated temperatures?

Unfortunately, this was a requirement by the importing authorities in New Zealand. We acknowledge that freeze-drying would have been optimal since it is the standard method for lipid biomarker extractions and that heating at higher temperature has been shown to potentially impact biomarker distributions. However, we look at predominantly ancient organic matter where initial diagenesis should also have had an impact on biomarker distributions, so we would expect only limited impact on the biomarker distributions from the drying procedure. According to the biomarker guide by Peters et al, for instance, the n-C12 alkane has a boiling point of 216°C, so we would not expect major differences due to the drying of the samples at 80°C. Also, we still detect ancient, intact plant sterols among other more labile biomarker suggesting that degradation by sample drying cannot have had a major impact on the robustness of our dataset. Indeed,

a bulk of literature shows that oven-drying does not alter significantly plant biomarkers (Suh & Diefendorf, 2020 and references therein). Therefore, we are confident that the drying process had limited impacts on the biomarker distribution and stable isotope composition.

• Also line 113: what is the evidence that this drying procedure didn't introduce contamination? Were blanks included?

We did not include blanks during the drying procedure, but we did comprehensive blank control during all steps of the biomarker extraction, processing and analysis, where we used large volumes of material. We did not find any plastic or any other contamination in our samples, so our samples and results are robust, with no evidence of being compromised in any way.

• Line 137: indicates that the polar fractions were derivatised with BSTFA, but later (line 158) the ions selected for the brGDGT analysis are given. Would they not have been impacted by the derivatisation?

Different aliquots were used. We made the clarification on lines 174 and 185.

• Line 170: this implies that odd numbered C chains are only created by diagenesis, but they are also found in plant waxes (as noted for section 3.4.2). Care needed here with phrasing.

Thank you for noting this. We changed it according to the recommendation made by reviewer 1: "Natural distributions or well-preserved *n*-alkane signatures are expected to show a predominance of odd-numbered carbon chains because of decarboxylation of fatty acids that show characteristic even carbon number predominance."

• Section 3.6: there is no mention of the RPO experiments (figure 7 shows them), so these need to be included.

We have merged sections 3.5 and 3.6 together and added details on the RPO experiments.

• Section 3.7: please check the readme file for chcluster, but I thought that this method explicitly included an acknowledgment of the ordering of the samples, so that the suggestion in line 240 that the statistics did the grouping (and found an anomalous sample) seems incorrect. Is the anomalous sample flagged as a 4th cluster, and was it statistically significant? The anomalous sample only stands out as being anomalous on figure 2.

The anomalous sample was part of a  $4^{th}$  cluster and therefore we removed it all together for ease of grouping. This sample had particularly low C content (within error range) to which all biomarker indices were normalized, making it plot as an outlier in all the graphs that are normalized to  $C_{org}$ . Therefore, it was excluded from the interpretation.

• Line 243 flags two additional statistical tests which were performed but I have struggled to find where the results of this analysis are presented or used to inform the interpretations.

Thank you for picking-up on this. We did these tests for initial screening. Since we do not present the results from these tests we removed the following statement: "An ANOVA within subjects and post hoc Tukey HSD tests were then conducted on the results to determine whether the means of each variable in the three units were statistically different."

• Line 251 is the only time that the cholesterol / plant wax ratio is referred to. What is this showing, and what is the reason for the peak in unit 2?

We removed this ratio based on a comment from reviewer 1.

---

## Author Response (AR2)

**Dear authors,**

thank you for a comprehensive and clear revision of your manuscript, which can now be accepted for publication in Biogeosciences. Based on your revised version I only have a minor, technical comments.

- Rephrase d13C-CaCO3 so that CaCO3 is not in subscript, for instance by using a dash after d13C. This way the chemical notation of CaCO3 can be written correctly (with subscript). Or, consider writen inorg, as an alternative for d13Corg.

**Done.**

- Appreciation for the effort of the reviewers can be reflected by a short statement in the acknowledgments.

**Done.**

Additional private note (visible to authors and reviewers only): Thanks to all reviewers and authors for constructive comments and replies.

Thank you to both reviewers and to the editor for their comments on the manuscript. We believe it has greatly increased the quality of the manuscript.